# Momentum-space signatures of Berry flux monopoles in the Weyl semimetal TaAs

M. Ünzelmann[1,10], H. Bentmann [1,10 ✉], T. Figgemeier[1,10], P. Eck[2,10], J. N. Neu[3,4], B. Geldiyev[1], F. Diekmann[5,6], S. Rohlf[5,6], J. Buck[5,6], M. Hoesch [7], M. Kalläne[5,6], K. Rossnagel [5,6,7], R. Thomale[2], T. Siegrist[3,4], G. Sangiovanni [2], D. Di Sante [2,8,9] & F. Reinert[1]

Since the early days of Dirac flux quantization, magnetic monopoles have been sought after as a potential corollary of quantized electric charge. As opposed to magnetic monopoles embedded into the theory of electromagnetism, Weyl semimetals (WSM) exhibit Berry flux monopoles in reciprocal parameter space. As a function of crystal momentum, such monopoles locate at the crossing point of spin-polarized bands forming the Weyl cone. Here, we report momentum-resolved spectroscopic signatures of Berry flux monopoles in TaAs as a paradigmatic WSM. We carried out angle-resolved photoelectron spectroscopy at bulk-sensitive soft X-ray energies (SX-ARPES) combined with photoelectron spin detection and circular dichroism. The experiments reveal large spin- and orbital-angular-momentum (SAM and OAM) polarizations of the Weyl-fermion states, resulting from the broken crystalline inversion symmetry in TaAs. Supported by first-principles calculations, our measurements image signatures of a topologically non-trivial winding of the OAM at the Weyl nodes and unveil a chirality-dependent SAM of the Weyl bands. Our results provide directly bulk-sensitive spectroscopic support for the non-trivial band topology in the WSM TaAs, promising to have profound implications for the study of quantum-geometric effects in solids.

[1] Experimentelle Physik VII and Würzburg-Dresden Cluster of Excellence ct.qmat, Universität Würzburg, Würzburg, Germany. [2] Theoretische Physik I, Universität Würzburg, Würzburg, Germany. [3] Department of Chemical and Biomedical Engineering, FAMU-FSU College of Engineering, Tallahassee, FL, USA. [4] National High Magnetic Field Laboratory, Tallahassee, FL, USA. [5] Institut für Experimentelle und Angewandte Physik, Christian-Albrechts-Universität zu Kiel, Kiel, Germany. [6] Ruprecht Haensel Laboratory, Kiel University and DESY, Kiel, Germany. [7] Deutsches Elektronen-Synchrotron DESY, Hamburg, Germany. [8] Department of Physics and Astronomy, University of Bologna, Bologna, Italy. [9] Center for Computational Quantum Physics, Flatiron Institute, New York, NY, USA. [10] These authors contributed equally: M. Ünzelmann, H. Bentmann, T. Figgemeier, P. Eck. ✉email: hendrik.bentmann@physik.uni-wuerzburg.de

Topological semimetals have become a fruitful platform for the discovery of quasiparticles that behave as massless relativistic fermions predicted in high-energy particle physics[1–6]. A prominent example are Weyl fermions, which are realized at topologically protected crossing points between spin-polarized electronic bands in the bulk band structure of non-centrosymmetric or ferromagnetic semimetals[1,2,7–11]. Near a Weyl node the momentum-resolved two-band Hamiltonian takes the form $H \propto \pm \boldsymbol{\sigma} \cdot \mathbf{k}$, giving rise to the linear energy–momentum dispersion relation of a massless quasiparticle[4]. The topological structure of the Weyl node, however, is not encoded in the energy spectrum, but rather manifests in the momentum-dependence of the eigenstates, i.e., the electronic wave functions. The pseudospin $\boldsymbol{\sigma}$ and the Berry curvature $\boldsymbol{\Omega}$ wind around the Weyl node, forming a Berry flux monopole in three-dimensional (3D) momentum space[12,13]. The nontrivial winding of $\boldsymbol{\sigma}$ stabilizes the Weyl node by a topological invariant, a nonzero Chern number of $C = \pm 1$ (ref. [4]).

Until today, angle-resolved photoelectron spectroscopy (ARPES) experiments have confirmed a number of materials as Weyl semimetals (WSM), based on a comparison of the measured bulk band structure to band calculations and the observation of surface Fermi arcs[1,9,10,14,15]. Manifestations of the nontrivial topology have also been found, accordingly, in magnetotransport experiments[16], by scanning tunneling microscopy[17], and via optically induced photocurrents[18]. The winding of the electronic wave functions in momentum space, however, which characterizes the immediate effect of a Berry flux monopole and thus the topology of the WSM, has so far remained elusive.

While the pseudospin $\boldsymbol{\sigma}$ for a Dirac-Hamiltonian universally shows nontrivial winding, the underlying microscopic degrees of freedom may vary from one system to another[19]. In two dimensions, examples include the sublattice degree of freedom for graphene and the spin-angular momentum (SAM) for the surface states in topological insulators (TI). Accordingly, the nontrivial Berry-phase properties have been addressed by quasiparticle interference STM imaging and dichroic ARPES in graphene[20,21] and by spin-resolved ARPES in TI[22]. Likewise in 3D WSM, one may expect the relevant degrees of freedom to depend on the considered material system. While previous theories have suggested orbital-sensitive dichroic effects in momentum-resolved spectroscopies, as a probe of topological characteristics in topological semimetals[23–25], such approaches have previously not reached application to experimental data.

In the present work, we find that the orbital-angular momentum (OAM) $\mathbf{L}$ plays a crucial role in the Weyl physics of the paradigmatic WSM TaAs and, like the pseudospin $\boldsymbol{\sigma}$, displays a topologically nontrivial winding at the Weyl points. Our experiments are based on soft X-ray (SX) ARPES, which allows for the systematic measurement of bulk band dispersions by virtue of an increased probing depth and excitation into photoelectron final states, with well-defined momentum along $k_z$ perpendicular to the surface[26], when compared to surface-sensitive ARPES experiments at VUV photon energies. SX-ARPES has been the key method to probe chiral-fermion dispersions in topological semimetals[1,2,5,27–29], but its combination with spin resolution (SR) and circular dichroism (CD) is challenging and not widely explored[30]. On the other hand, at lower excitation energies SR is routinely used to detect the SAM of electronic states[22] and CD has been introduced as a way to address the OAM[31–35]. We performed SX-ARPES measurements combined with SR and CD to probe the SAM and OAM in the bulk electronic structure of TaAs.

## Results

**Bulk band structure of TaAs.** TaAs crystallizes in the non-centrosymmetric space group $I4_1md$, as shown in Fig. 1a. According to the results of first-principles calculations, its bulk band structure features 12 pairs of Weyl points in the Brillouin zone, which divide into two inequivalent sets, $W_1$ and $W_2$ (refs. [7,8]). Here, we focus on the $W_2$ points which are located in $k_x$–$k_y$ planes at $k_z = \pm 0.59\,\frac{2\pi}{c}$, with the length $c$ along $z$ denoting the conventional unit cell. In agreement with earlier works[1,2], our SX-ARPES data in Fig. 1 predominantly reflect the bulk bands of TaAs, allowing us to address the bulk band structure through variation of the photon energy. The experimental dispersions along the $\Gamma\Sigma$ and $ZS$ high-symmetry lines compare well with the calculated bands (Fig. 1c–d). We further performed $h\nu$-dependent measurements to trace the band dispersion along $k_z$, for which we likewise achieve agreement with theory (Fig. 1e and Supplementary Note 1). Based on this, we determine a photon energy of approximately $h\nu = 590$ eV that allows us to reach a final-state

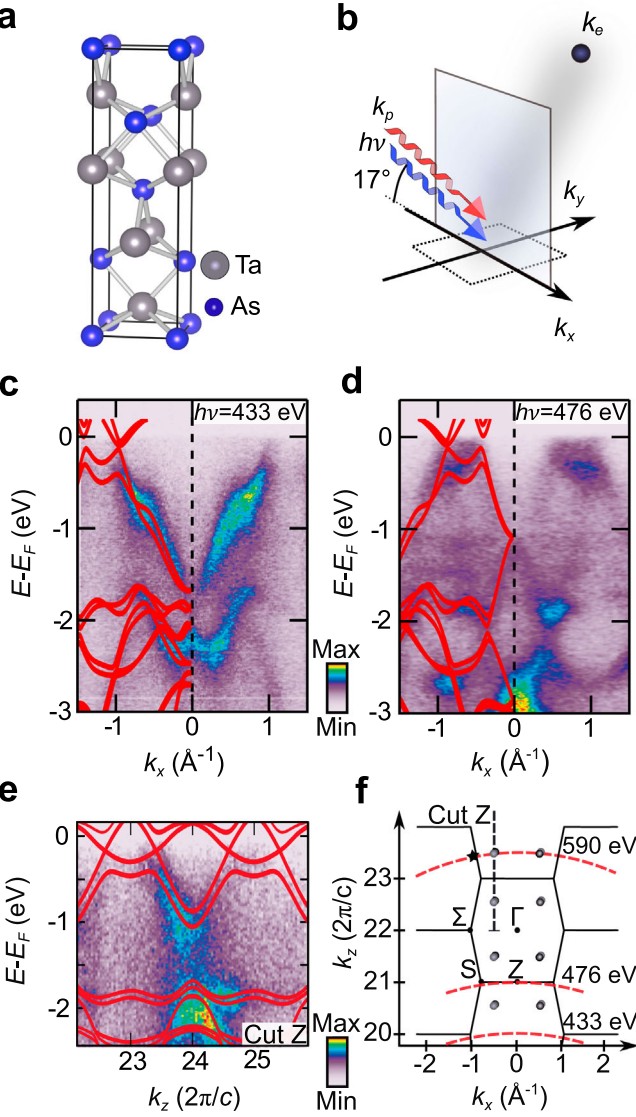

**Fig. 1 Bulk electronic structure of TaAs. a** Bulk crystal structure of TaAs with space group $I4_1md$. **b** Experimental geometry of the angle-resolved photoemission spectroscopy (ARPES) experiment using circularly polarized soft X-ray radiation. **c, d** ARPES data sets along the $\Gamma\Sigma$ and $ZS$ high-symmetry directions of the bulk Brillouin zone. **e** ARPES data set along $k_z$ at $k_x = -0.53$ Å$^{-1}$, corresponding to the calculated $k_x$ position of the $W_2$ Weyl nodes (cut $Z$ in **f**). In **c–e**, the red lines represent the calculated band dispersion. **f** Bulk Brillouin zone structure in the $k_x$–$k_z$ plane. $W_2$ Weyl nodes at $k_z = 23.41\,\frac{2\pi}{c}$ are addressed at a photon energy of approximately $h\nu = 590$ eV (cf. Figs. 2 and 3). The star marks the $(k_x, k_z)$ position of the data set in Fig. 2a.

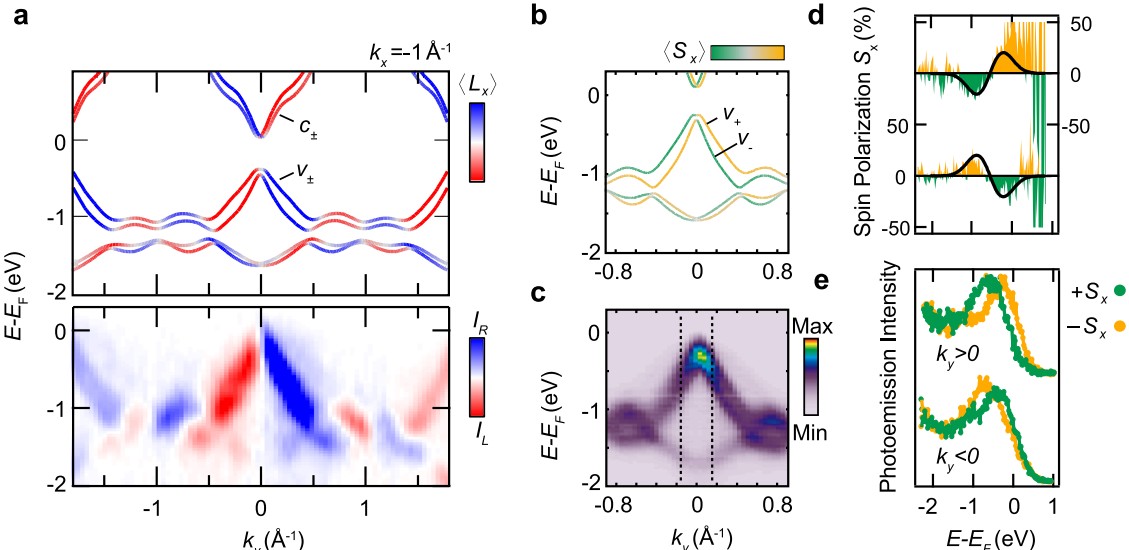

**Fig. 2 Orbital and spin-angular momentum of bulk electronic states in TaAs. a** Calculated bulk band dispersion of TaAs projected on the component $L_x$ of the orbital-angular momentum (OAM). The cut along $k_y$ is taken at $k_z = -0.59\frac{2\pi}{c}$ and $k_x = -1\,\text{Å}^{-1}$. The spin–orbit split valence band $v_\pm$ and conduction band $c_\pm$ are indicated. The calculation is compared to a corresponding CD-ARPES data set in the bottom panel, obtained at $h\nu = 590$ eV (see indication in Fig. 1f). The red/blue color code represents the circular dichroism (CD) measured with circularly polarized X-rays incident in the $xz$-plane (cf. Fig. 1b). **b** Calculated band dispersion projected on the component $S_x$ of the spin-angular momentum (SAM), showing opposite $S_x$ for $v_\pm$. **c** ARPES data set along $k_y$ taken at $h\nu = 590$ eV and $k_x = -1\,\text{Å}^{-1}$. Dashed lines indicate the $k_y$ positions of the spin-resolved energy distribution curves (EDC) in **e**. **d**, **e**, Spin-resolved EDC and measured spin polarization at $\pm k_y$, as indicated in **c**. The spin quantization axis is along $x$. Data sets at $+k_y$ ($-k_y$) were measured with left (right) circularly polarized light (see Supplementary Note 2 for details).

momentum corresponding to the $W_2$ Weyl points within experimental uncertainty (Fig. 1e, f).

**Orbital and spin-angular momentum of the bulk states.** The formation of Weyl nodes in TaAs relies on the coexistence of inversion-symmetry breaking (ISB) and spin–orbit coupling (SOC)[7,8], which induces a spin splitting into nondegenerate bands. The results of our first-principles calculations in Fig. 2a, b show this splitting for the bulk valence and conduction bands, which split up into branches $v_+$ and $v_-$, as well as $c_+$ and $c_-$, respectively. Note that the crossing points between the upper valence band $v_+$ and the lower conduction band $c_-$ define the Weyl nodes in TaAs (see below). Going beyond the band dispersion, we explore the impact of ISB and SOC on the electronic wave functions. According to our calculations, the key consequence of ISB is the formation of a sizable OAM **L** in the Bloch wave functions (Fig. 2a and Supplementary Fig. 3). The spin-split branches in the valence and conduction band carry parallel OAM, while the SAM is antiparallel (Fig. 2a, b). This indicates a scenario for the bulk states in TaAs, where the energy scale associated with ISB dominates over the one of SOC[32,33,36] (see also Supplementary Fig. 4). Moreover, our calculations show that $v_\pm$ and $c_\pm$ carry opposite OAM, indicating that the Weyl nodes are crossing points between bands of opposite OAM polarization.

To confirm the formation of OAM experimentally, we measured the CD signal, i.e., the difference in photoemission intensity for excitation with right and left circularly polarized light, which has been shown to be approximately proportional to the projection of **L** on the light propagation direction $\mathbf{k}_p$ (refs. [31,32]). The grazing light incidence in the $xz$-plane of our experimental geometry (Fig. 1b), thus implies that the measurements predominantly reflect the $L_x$ component of the OAM. Note that the relation between CD and OAM, derived in refs. [31,32], relies on the free-electron final-state approximation, which can be expected to hold particularly well at the high excitation energies used in the present experiment. Indeed, a comparison of the measured CD to the $L_x$-projected band structure shows a remarkable agreement over wide regions in momentum space, as seen in Figs. 2a and 3a, b. The detailed match between experimental data and theory proves that the measured CD indeed closely reflects the momentum-resolved local OAM of the bulk states in TaAs. As expected theoretically[31,32], the OAM of the initial state is thus the most important source of CD under the present experimental conditions as opposed to mere geometric or final-state effects. Particularly the latter can become relevant at lower excitation energies, where the final state is not well-approximated in the free-electron picture[37,38].

While OAM is induced by ISB already in the absence of SOC, the presence of Weyl nodes requires a SOC-induced formation of spin polarization[7,8]. An experimental proof of spin-polarized bulk states in TaAs and related transition-metal monopnictides has remained elusive up to now. Our spin-resolved SX-ARPES measurements directly verify the predicted SAM of the bands $v_\pm$. They confirm an opposite $S_x$ for $v_+$ and $v_-$ (Fig. 2b–e), while our CD-ARPES data prove a parallel alignment of $L_x$, in line with our calculations (Fig. 2a). Moreover, the spin-resolved data confirm an opposite sign of $S_x$ at $+k_y$ and $-k_y$. Overall, our experiments establish sizable OAM and SAM polarizations of the bands forming the Weyl nodes in TaAs.

**Circular dichroism and OAM near the Weyl nodes.** To explore the momentum-dependence of the OAM in more detail, we consider momentum distributions of the measured CD and the calculated $L_x$ for the band $v_\pm$ in an equi-$k_z$ plane of the $W_2$ nodes (Fig. 3a, b). Besides the good agreement of experiment and theory, it is evident from the data that the Weyl-node pair near $k_x = -0.5\,\text{Å}^{-1}$, marked in Fig. 3b, is located at a distinctive position within the OAM texture. A quantitative comparison of the CD and the calculated OAM along $k_x$ paths through the two Weyl nodes of opposite chirality is shown in Fig. 3c. The experiment reveals sign changes of $L_x$ close to the Weyl nodes and an opposite overall sign of $L_x$ for the two nodes, consistent with the

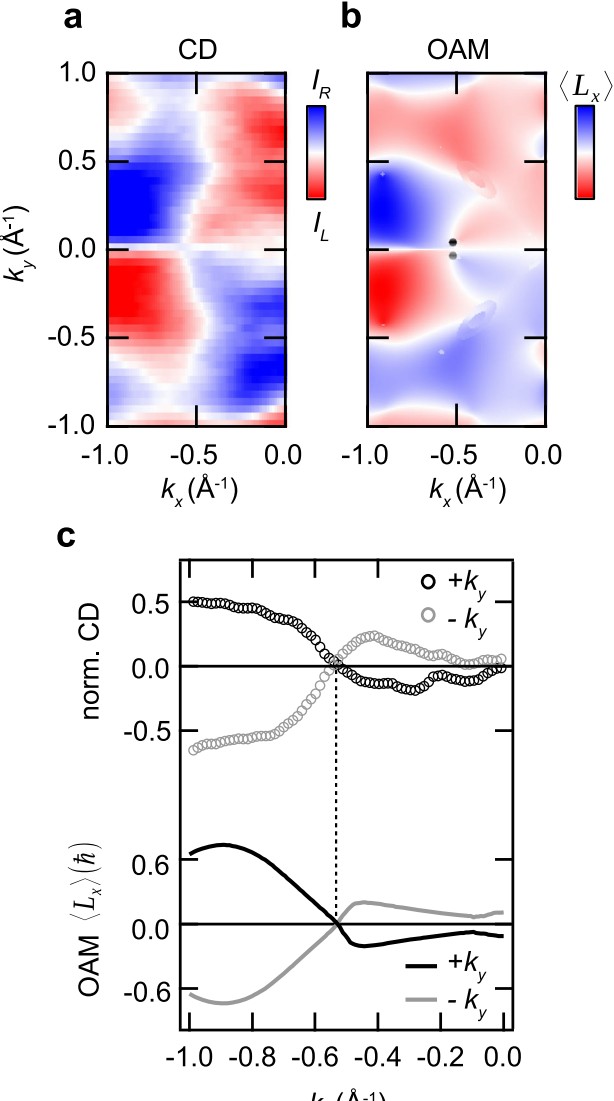

**Fig. 3 Momentum-space texture of circular dichroism and orbital-angular momentum. a, b** Momentum distributions of measured circular dichroism (CD) and calculated $L_x$ component of the orbital-angular momentum (OAM) at $k_z = -0.59\frac{2\pi}{c}$, corresponding to the $W_2$ Weyl nodes. The data sets were obtained by integrating over an energy range from 0 eV to $-1.2$ eV, the width of the bands $v_\pm$ (cf. Fig. 2a). The calculated positions of the $W_2$ Weyl nodes are indicated in **b**. The CD-ARPES data were acquired at $h\nu = 590$ eV. **c** CD signal normalized to the total photoemission intensity and $L_x$ component of the OAM along $k_x$ cuts through Weyl nodes of opposite chirality. Sign changes are observed near the Weyl nodes.

calculated OAM. Although the data in Fig. 3 reflects information integrated over both bands $v_+$ and $v_-$, these observations suggest a role of the OAM in the Weyl physics of TaAs.

To examine the OAM and CD of the Weyl cone, we focus on the band dispersion along $k_x$ across a $W_2$ Weyl node in Fig. 4a–e. For the band $v_+$, which forms the lower part of the Weyl cone, the two branches on the left and on the right side of the Weyl node are selectively probed by the circular light polarization: using right circularly polarized light (Fig. 4d), there is a high photoemission intensity for the left branch of $v_+$, while the right branch is suppressed. Vice versa, for left circularly polarized light, we find the left branch to be suppressed, while the right branch is more strongly excited (Fig. 4e). Accordingly, the band $v_+$, shows

an inversion of OAM across the Weyl node (Fig. 4b). This behavior is supported by a quantitative analysis of the intensities, shown in Fig. 4c. These observations indicate an opposite OAM for the upward and downward dispersing branches of $v_+$ and thus an OAM sign change across the Weyl node, in agreement with our calculations. It is noteworthy that the magnitude of the CD in general does not reach 100%, as seen, e.g., from the analysis in Fig. 4c.

The fact that the Weyl node is located slightly below the Fermi level further allows us to estimate the relative OAM orientation of the lower and the upper part of the Weyl cone. In Fig. 4h, we consider energy distribution curves (EDC) at a wave vector $k_x$ close to the Weyl node. A three-peak structure is observed in the EDC and attributed to the bands $v_-$, $v_+$, and $c_-$. The CD for the bands $v_+$ and $c_-$ is opposite confirming the opposite OAM of these bands predicted by our calculations (Fig. 4b–e). Note that for left circularly polarized light the intensity of the band $v_+$ drops toward the Fermi energy, which, in combination with finite linewidth broadening, gives rise to some deviations between the CD signal and calculated OAM in Fig. 4b (see also Supplementary Figs. 10–12 for a more detailed discussion). In particular, unlike for the band $v_+$, our present data does not allow us to discern a CD reversal of the band $c_-$ across the Weyl point close to the Fermi level.

Our measurements and calculations in Fig. 4 directly image characteristic modulations of the Bloch wave functions near the $W_2$ Weyl nodes. Further evidence for momentum-dependent changes also of the $L_y$ component near the $W_2$ nodes is presented in Supplementary Fig. 9. It is important to note that the observed sign reversal of $L_x$ across the Weyl node is not enforced by any symmetry. Rather, it reflects a true band-structure effect that characterizes the Weyl-fermion wave functions, namely the crossing of two bands carrying opposite OAM (Fig. 4b–e). This situation is different from the surface states in TI[22,31] or related examples[39,40], where a sign reversal across the nodal point is strictly imposed by time-reversal or crystalline symmetries. From an experimental point of view, this makes the presently observed sign reversal of $L_x$ more forceful, as it is a band-structure-specific and not a symmetry-imposed texture change.

**Topological winding of OAM and Berry curvature.** Our results reveal a close correspondence between measured CD and calculated OAM. In this work, the OAM is computed in the local limit by projecting the Bloch wave function onto atom-centered spherical harmonics, i.e., we consider the quantum-mechanical expectation value of the atomic $L$ angular momentum operator. This approach offers a viewpoint complementary to the OAM given by the modern theory of polarization[41]. The fact that the CD rather corresponds to the local OAM[31,32] could be related to the local nature of the photoemission process itself[42].

The formation of OAM, which we observe here, provides direct evidence for the influence of ISB on the bulk Bloch states and, thus, suggests that these states also carry finite Berry curvature $\mathbf{\Omega}$, which likewise originates from ISB. In Fig. 4f, we consider momentum distributions of CD and $L_x$ for the band $v_+$, revealing a characteristic sign change along the contour around the Weyl nodes. The calculated momentum texture of the $\Omega_x$-component of the Berry curvature qualitatively resembles the characteristics of the OAM and the measured CD (Fig. 4f), suggesting that these quantities reflect the topologically nontrivial winding of the wave functions near the Weyl nodes[21,34,35,43]. Indeed, our calculations show explicitly that for TaAs the nontrivial topology of the Berry flux monopoles is encoded in the momentum-dependence of the OAM, while the SAM shows topologically trivial behavior. The topological nature of the field configuration is determined by the

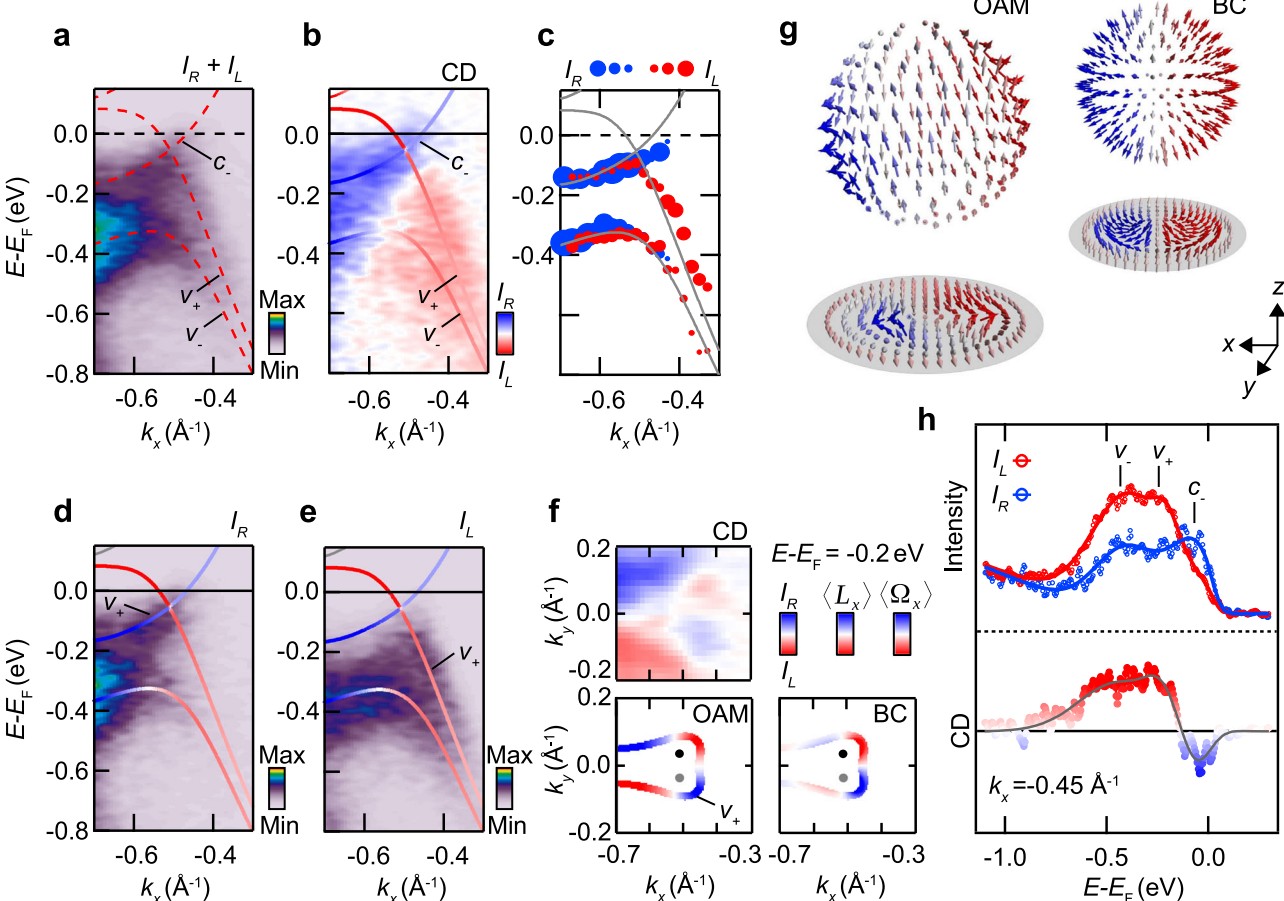

**Fig. 4 Topological winding and orbital-angular momentum of Weyl-fermion states. a–e** ARPES data sets and calculated $L_x$-projected band dispersion along a $k_x$ cut through a $W_2$ Weyl node at $k_y = 0.037$ Å$^{-1}$ and $k_z = -0.59 \frac{2\pi}{c}$. The measurements were taken with left $I_R$ (**d**) and right $I_L$ (**e**) circularly polarized light. Panel **a** shows the sum and **b** the difference (circular dichroism, CD) of $I_L$ and $I_R$. Panel **c** shows the results of Gaussian peak fits to energy distribution curves (EDC) in the data sets $I_R$ (blue) and $I_L$ (red) for different $k_x$ (cf. Fig. S11). The dot size represents the extracted peak intensity. **f** Momentum distributions of CD, $L_x$, and the $\Omega_x$ component of the Berry curvature (BC) in the vicinity of the $W_2$ Weyl nodes. The data sets were obtained by integrating over an energy range from −0.16 eV to −0.24 eV. **g** OAM and Berry curvature calculated on a small sphere around a $W_2$ Weyl node and the corresponding azimuthal equidistant projections. Both textures are characterized by a nontrivial Pontryagin index of $S = 1$. **h** EDC of $I_L$ and $I_R$ (upper panel) and the corresponding CD (lower panel) taken at $k_x = -0.45$ Å$^{-1}$. The three peaks in the EDC are assigned to the bands $v_+$, $v_-$, and $c_-$.

Pontryagin index calculated on a sphere surrounding the Weyl point

$$S = \frac{1}{4\pi} \int \mathbf{n} \cdot \left[ \frac{\partial \mathbf{n}}{\partial x} \times \frac{\partial \mathbf{n}}{\partial y} \right] dx dy \qquad (1)$$

where $\mathbf{n}$ is the unit vector of the field and the integral is taken over the 2D manifold covering the sphere. $S$ is proportional to the Berry phase in momentum space accumulated by electrons encircling the Weyl point. In our calculations, the Pontryagin index of the vector fields of Berry curvature and OAM is $S = 1$ (Fig. 4g), while the SAM has $S = 0$ (Supplementary Fig. 8). Analogously to the Berry curvature, the Pontryagin index of the OAM is nonvanishing only if the sphere surrounds a Weyl node, and changes sign when the other Weyl node of the pair is considered. The Berry flux monopoles in TaAs thus derive from the orbital degrees of freedom of the electronic wave functions, and their topology manifests in a nontrivial texture winding of the OAM.

We indeed theoretically verified that in the WSM TaP[44] and LaAlGe[45] the Pontryagin index of the OAM vector field winds around the type-I Weyl nodes. Interestingly, LaAlGe hosts two type-I and one type-II Weyl dispersions[46]. Our analysis brings further evidence of a nontrivial winding of OAM only at type-I

Weyl cones (see also Supplementary Note 5), raising intriguing questions about the underlying conditions of spectroscopic manifestations of Berry flux monopoles. Preliminary investigations suggest that non-symmorphic symmetries and a large atomic-angular momentum ($d$ orbitals involved in the low-energy description) may play a decisive role. Irrespectively, our work shows that the OAM can constitute—though not universally—an experimentally accessible quantity that reflects the nontrivial band topology in a WSM.

**Energy hierarchy of ISB and SOC.** For two-dimensional systems, the formation of OAM due to ISB has been shown to be the microscopic origin of SOC-induced spin splittings[36,47]. The present experiments establish a first instance where OAM-carrying bands are observed in a 3D bulk system. Our observations of SAM and OAM in TaAs indicate that an OAM-based origin of the spin splitting also applies to bulk systems with ISB and, as such, underlies the WSM state at the microscopic level. Specifically, the relative alignments of OAM and SAM in the bands $v_\pm$, determined from our CD-ARPES and spin-resolved data (cf. Fig. 2), imply an energy hierarchy of the bulk band structure in which ISB dominates over SOC[32,33]. In this case, the electronic states can be classified in terms of their OAM, as SOC

does not significantly mix states of opposite OAM polarization. The reversal of $L_x$, that we observe at the Weyl node, thus indicates an orbital-symmetry inversion accompanying the crossing of valence and conduction bands in TaAs. This supports an earlier theory predicting that the Weyl-semimetal state in TaAs originates from an inversion between bands of disparate orbital symmetries[7].

## Discussion

Previous SX-ARPES experiments succeeded in measuring the bulk band dispersion in WSM and other topological semimetals[1,2,5,6,28,29,48]. The present experiment for TaAs goes beyond the dispersion and probes the spin–orbital-components of the wave functions, which encode the topological properties. Our measurements unveil a characteristic reversal in the OAM texture of the bulk Weyl-fermion states at momenta consistent with the termination points of the surface Fermi arcs in TaAs[1,9,48]. Together, this constitutes a remarkably explicit spectroscopic manifestation of bulk-boundary correspondence in a WSM. In this regard, our results push forward the use of circularly polarized light as a probe of Weyl-fermion chirality to the momentum-resolved domain[18].

Our results open a pathway to probe SAM and OAM textures in the bulk band structures of recently discovered magnetic WSM with broken time-reversal symmetry[10,11], and of other topological semimetals hosting chiral fermions of higher effective pseudospin and higher Chern numbers, and thus with topological properties distinct from Weyl fermions[5,6,28,29,49]. This may allow to address the topological features in these band structures. For example, DFT calculations indicate a topological winding of the SAM at the Weyl nodes in the magnetic WSM $HgCr_2Se_4$ (ref. [50]). More broadly, our results establish a possibility of probing the topological electronic properties of a condensed matter system directly from a bulk perspective without resorting to the corresponding boundary modes.

## Methods

**Experimental details.** We carried out SX angle-resolved photoemission spectroscopy (SX-ARPES) experiments at the ASPHERE III endstation at the Variable Polarization XUV Beamline P04 of the PETRA III storage ring at DESY (Hamburg, Germany). SX photons with a high degree of circular polarization impinge on the sample surface under an angle of incidence of $\alpha \approx 17°$ (cf. Fig. 2b). TaAs single crystals were cleaved in situ with a top-post at a temperature <100 K and a pressure better than $5 \times 10^{-9}$ mbar. ARPES data were were collected using a SCIENTA DA30-L analyzer at a sample temperature of ~50 K and in ultra-high vacuum <3 × $10^{-10}$ mbar. The angle and energy resolution of the ARPES measurements was ca. $\Delta\theta \approx 0.1°$ and $\Delta E \approx 50$ meV. Spin-resolved ARPES was performed by use of a Mott detector (Scienta Omicron). The Au scattering target had an effective Sherman function of 0.1. The energy resolution was ca. $\Delta E \approx 600$ meV, while the angle resolution was $\Delta\theta \approx 6°$ in $k_x$ direction. The CD-ARPES and spin-resolved ARPES data were obtained using the deflection mode of the spectrometer, so that the experimental geometry stays fixed during the data acquisition procedures.

The growth of single crystals was performed via chemical vapor transport reactions. Sealed silica ampoules with inner diameter of 14 mm and length of 10 cm were loaded with Ta foil and As chunks in a stoichiometric ratio so that the total mass of the reactants was 1.2 g. After adding 0.055 g $I_2$ as transport agent, the ampoules were sealed airtight under a vacuum of $\approx 10^{-5}$ mTorr. Then they were loaded into a single zone tube furnace with a defined temperature gradient across the length of the tube. The angle of the crucibles against the horizontal was $\approx 20°$. The temperature was increased from room temperature up to 640 °C at 5 K h$^{-1}$, where it was held constant for 24 h and subsequently heated to 1000 °C at a rate of 2.5 K h$^{-1}$. Crystal growth proceeded over the following 3 weeks in a temperature gradient of 1000 °C: 950 °C (ref. [51]). After cooling radiatively to room temperature, TaAs single crystals with a volume up to 2 mm³ were obtained. The crystal structure and stoichiometry were confirmed using single-crystal X-ray diffraction[52] and energy dispersive X-ray spectroscopy methods (Zeiss 1540 XB Crossbeam Scanning Electron Microscope).

**Theoretical details.** We consider in our theoretical study the non-centrosymmetric primitive unit cell of TaAs (space group $I4_1md$) with a lattice constant of 6.355 Å. We employ state-of-the-art first-principles calculations based on the density functional theory as implemented in the Vienna ab initio simulation

package (VASP)[53], within the projector-augmented plane-wave (PAW) method[54,55]. The generalized gradient approximation as parametrized by the PBE-GGA functional for the exchange-correlation potential is used[56] by expanding the Kohn–Sham wave functions into plane waves up to an energy cutoff of 400 eV. We sample the Brillouin zone on an $8 \times 8 \times 8$ regular mesh by including SOC self-consistently[57]. For the calculation of the OAM, the Kohn–Sham wave functions were projected onto a Ta $s$, $p$, $d$-, and As $s$, $p$-type tesseral harmonics basis as implemented in the WANNIER90 suite[58]. The OAM expectation values were then obtained in the atom-centered approximation by rotating the tesseral harmonics basis into the eigenbasis of the OAM-operator, i.e., the spherical harmonics.

## Data availability
The data that support the findings of this study are available from the corresponding author upon reasonable request.

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

## Acknowledgements

This work is funded by the Deutsche Forschungsgemeinschaft (DFG, German Research Foundation) through Project-ID 258499086-SFB 1170 (projects A01 and C07), the Würzburg-Dresden Cluster of Excellence on Complexity and Topology in Quantum Matter −ct.qmat Project-ID 390858490-EXC 2147, and RE1469/13-1. We acknowledge DESY (Hamburg, Germany), a member of the Helmholtz Association HGF, for the provision of experimental facilities. Parts of this research were carried out at PETRA III and we would like to thank Kai Bagschik, Jens Viefhaus, Frank Scholz, Jörn Seltmann, and Florian Trinter for assistance in using beamline P04. Funding for the photoemission spectroscopy instrument at beamline P04 (Contracts 05KS7FK2, 05K10FK1, 05K12FK1, and 05K13FK1 with Kiel University; 05KS7WW1 and 05K10WW2 with Würzburg University) by the Federal Ministry of Education and Research (BMBF) is gratefully acknowledged. The research leading to these results has received funding from the European Union's Horizon 2020 research and innovation programme under the Marie Skłodowska-Curie Grant Agreement No. 897276. We gratefully acknowledge the Gauss Centre for Supercomputing e.V. (www.gauss-centre.eu) for funding this project by providing computing time on the GCS Supercomputer SuperMUC at Leibniz Supercomputing Centre (www.lrz.de). J.N.N. and T.S. acknowledge support from the National Research Foundation, under Grant No. NSF DMR-1606952. The crystal synthesis and characterization was carried out at the National High Magnetic Field Laboratory, which is supported by the National Science Foundation, Division of Materials Research under Grant No. DMR-1644779 and the state of Florida. This publication was supported by the Open Access Publication Fund of the University of Würzburg.

## Author contributions

M.Ü. and T.F. performed the experiments, with support from H.B., B.G., F.D., S.R., J.B., M.H., M.K., and K.R. M.Ü. and T.F. analyzed the experimental data. P.E. and D.D.S. performed the first-principles and model calculations. J.N.N., T.F., and T.S. synthesized and characterized the TaAs samples. All authors contributed to the interpretation and the discussion of the results. H.B. wrote the manuscript with contributions from M.Ü., T.F., P.E., R.T., D.D.S., G.S., and F.R. H.B. conceived and planned the project.

## Funding

## Competing interests

The authors declare no competing interests.
