## [Peer Review File · Nature Communications]

REVIEWER COMMENTS

Reviewer #2 (Remarks to the Author):

In the previous round, I have summarized the work of Uenzelmann et al. and assessed the relevance of the work for the scientific community. In short, I believe that if the authors can provide strong enough evidence for their conclusions, I think that their work will be novel and of high importance for scientists in the field of topological matter science and related disciplines, because it would establish an alternative and even more direct method to detect the presence of Berry curvature monopoles in solids. This would be of high impact especially if they can show that this method is applicable to other topological (semi-) metals beyond TaAs.

Whilst I remain excited about the authors' claims and appreciate the theoretical clarifications in their rebuttal letter, I am afraid that I still have substantial reservations about the quality of the evidence that is justifying their conclusions, which need to be resolved before I can recommend publication in Nature Communications.

In summary, I am not convinced yet that the experimental circular dichroism (CD) data presented here provides strong evidence for detecting the predicted $\langle L \rangle$ OAM monopole in TaAs, and I am concerned that some of the presentations of the data (particularly the checkerboard patterns) may be misleading for some readers. Moreover, from what I learned from the rebuttal letter, it seems that the predicted signature of the $\langle L \rangle$ OAM monopole is not connected to a Berry curvature monopole in general, but it appears to be a particular feature of TaAs that these two quantities coincide close to the Weyl-point. According to the calculations mentioned in the rebuttal letter, this (accidental?) coincidence also appears for some other but not all Weyl-points. When reading the title, abstract, introduction and discussion section, I am under the impression that the $\langle L \rangle$ OAM monopole is a universal feature of Berry curvature monopoles, which does not seem to be true. These sections should be revised to clarify this point for the reader.

Here is a detailed description of my concerns:

(1) Comments on the interpretation of the experimental data

Assuming for the time being that the calculated texture $\langle L \rangle$ OAM does reflect the texture of the Berry curvature, the experimental data presented here does not seem to agree with what is expected from the ab-initio calculations for the Berry curvature monopole:

a. The ab-initio calculations in Fig. 4b predict that the Berry curvature monopole consists of a band crossing of two bands with opposite OAM. This implies that, firstly, (i) at the Weyl-crossing point, there should be vanishing OAM, and secondly, (ii) to the left and to the right of Weyl-point, the upper and the lower branch of the cone should have opposite OAM. The circular dichroism (CD) data presented here does not show any signs of (i) vanishing OAM at the predicted crossing point, instead, the vicinity of the Weyl-point is completely blue with no apparent modulation of the CD-intensity. This fact contradicts the claim that a Berry curvature monopole was observed. The authors seem to imply in their rebuttal letter that this is due to momentum-dependent cross-section variations. However, I expect that the authors are aware that such matrix element effects can be suppressed in neighbouring Brillouin zones or by changing the photon energy, so unless the authors can produce such data, I am afraid that I do not find the current presentation very convincing.

The authors stress that they can show one-half of what is required for (ii), namely that the CD

switches sign for an EDC line cut on the right-hand side of the predicted position of the cone at $k_x=0.45 \text{ \AA}^{-1}$. However, this sign change is a necessary but not sufficient condition for the existence of the Weyl-point, and the inconsistencies described above seem to contradict its existence based on the data presented here.

One could imagine to shift the calculation a little bit to the right and a bit further down to align the Weyl-point with the white area between the red and blue regions. However, in that case it is unclear why there would be no signature of the red band reappearing close to the Fermi-level on the left side of the Weyl crossing, because then the energy resolution of 50 meV (as stated in the methods section) should be sufficient to resolve the splitting between the lower and the upper branch of the cone.

If the authors disagree with this assessment, it would be helpful to produce a simulated CD intensity distribution based on the ab-initio calculation that is taking into account Gaussian energy and momentum broadening and compare it to the measured CD data, rather than just plotting spaghetti plots on top of the data. I expect that such analysis would make it obvious that the experimental data does not agree very well with what is expected from the calculations.

b. In the main text of the manuscript and in the rebuttal letter, the authors imply that the experimentally observed and simulated checkerboard patterns in Figs. 3a-b and 4e are a signature of the Weyl-points in momentum space. As I have pointed out in the previous round of reviews - a point that has been ignored in the authors' rebuttal letter - such a comparison appears to be misleading the reader. This is because of the large energy integration window that was used to produce these plots, which includes both trivial bands that are not forming a Weyl cone (v^-) as well as bands forming a Weyl cone (v^+, c^-). However, because the trivial v^- bands seem to have accidentally almost the same OAM texture as the bands involved in the Weyl-cone (see calculations in Fig. 4b), the large integration window may produce a checkerboard pattern that is not due to the Weyl bands but due to the contributions from the trivial v^- bands. A more suitable analysis would involve an integration window that is limited to Weyl bands and not any other bands.

Moreover, as we can see from the OAM texture of the trivial v^- band, it appears that a trivial band not forming a Weyl-point may also abruptly change the sign of its CD- intensity at a non-high symmetry point, and may therefore produce a similar checkerboard pattern. The observation of a checkerboard pattern is therefore a necessary but no sufficient condition for detecting a Berry curvature monopole. What would be more convincing is to compare iso-energy surfaces above and below the Weyl-point energy - I would expect that these should show an inverted checkerboard pattern (with respect to each other) if the Weyl-point consists of a crossing of two bands with opposite OAM.

(2) Comments on the interpretation of the theoretical results

I appreciate that the authors have in their rebuttal letter clarified the difference between the OAM as conventionally defined in the literature and the quantity $\langle L \rangle$ that they also call OAM and which they use in all of their calculations presented in all of the figures of the manuscript. I believe that this distinction is absolutely crucial for the reader and needs to be mentioned in the discussion section because it is not yet clear that the quantity $\langle L \rangle$ OAM is really related to the Berry curvature for Weyl-semimetals in general, as is claimed in the first sentence of the discussion section.

What I find concerning is that the authors state in their rebuttal letter that there are many examples of other Weyl-semimetals with both type-I and type-II Weyl points that do not seem to show any non-trivial winding of the quantity $\langle L \rangle$ OAM. This seems to imply that what the authors try to probe in the experiment is actually not the intrinsic Berry curvature monopole, but a quantity that is (accidentally?) equivalent to the Berry curvature in the specific compound TaAs, but not in other compounds. I would

therefore strongly recommend that the authors should modify their title to make clear that their claims are specific to TaAs and not necessarily generalizable to other Weyl-semimetals or even other topological materials, such as those hosting higher-order fermions. They should also to disclose in the discussion section that there are some Weyl-points that only show a trivial winding of the quantity $\langle L \rangle$ OAM, despite being Berry curvature monopoles. They should also comment (and show at least in the supplementary materials) if for those Weyl-points the band crossing is made up of two bands with opposite $\langle L \rangle$ OAM. This is not clear from the discussion section in its current state.

Reviewer #3 (Remarks to the Author):

The true experimental proof of a topological insulator should be based on measurements of its bulk properties such as the winding number. Some theoretical proposals have been made to directly measure the winding number (for example, PRB 92, 195144 (2015)), but, to my best knowledge, it has not been realized. It is the reason why people resort to detection of surface states, which should be regarded as an indirect evidence in a strict sense. It is one of the reasons why theory usually takes the leading role in studies of topological materials – experiments provide supporting evidences.

The story goes similar for Weyl semiconductors; we usually look for Fermi arcs. However, a more direct evidence should be observation chiral structure of bulk Weyl fermion states and other bulk properties. It is what Unzelmann et al did in this work using bulk sensitive CD and spin-resolved ARPES for the first time. This scheme therefore can be applied to other materials in the future to establish Weyl fermion states of materials. More importantly, it will be useful in studying materials for which band calculations do not work well, i.e., correlated materials. In that case, experiments could take the leading role. The importance of this aspect is something I did not appreciate enough in the first review.

The manuscript also has useful discussion on the energy hierarchy and clearly showed OAM-based origin of the spin splitting in which the leading energy terms is determined by the OAM state rather than spin. This is the first experimental proof for such energy hierarchy in a 3D bulk system and provides a new microscopic insight on the Weyl physics.

There are some comments/questions on the replies to my previous specific questions. However, the answers to my questions/comments should not alter the main conclusion. As such, I fully support publication of the manuscript in Nature Communications.

Here are my comments to replies (to my specific comments in the first round) for the authors to consider.

1. The crystalline mirror plane is set as the experimental mirror plane as noted the Supplementary. Does that play a role at all in making the geometric effect negligible? Or is it solely from the fact that the final state is close to a free electron state? (Normally, the best experimental geometry would be setting the sample orientation so that the Weyl point falls on the experimental mirror plane since the geometrical contribution should be zero on the experimental mirror plane. However, this does not work in this case as OAM (L_x) also has odd distribution about the mirror plane.)
2. I agree with the authors that the final state is very close to a free electron state. This is no thorough experimental/theoretical proof that final state effect is negligible with a free electron final state in spite of mounting signs in experimental results. It would be important to concretely prove it in

order to establish SX CD-ARPES as a reliable tool to investigate OAM. I believe the added sentences are appropriate to describe the current situation.

AUTHORS' REPLY TO THE POINTS OF REVIEWER 2

A1 Reviewer 2: “In the previous round, I have summarized the work of Uenzelmann et al. and assessed the relevance of the work for the scientific community. In short, I believe that if the authors can provide strong enough evidence for their conclusions, I think that their work will be novel and of high importance for scientists in the field of topological matter science and related disciplines, because it would establish an alternative and even more direct method to detect the presence of Berry curvature monopoles in solids. This would be of high impact especially if they can show that this method is applicable to other topological (semi-) metals beyond TaAs. Whilst I remain excited about the authors’ claims and appreciate the theoretical clarifications in their rebuttal letter, I am afraid that I still have substantial reservations about the quality of the evidence that is justifying their conclusions, which need to be resolved before I can recommend publication in Nature Communications.

In summary, I am not convinced yet that the experimental circular dichroism (CD) data presented here provides strong evidence for detecting the predicted $\langle L \rangle$ OAM monopole in TaAs, and I am concerned that some of the presentations of the data (particularly the checkerboard patterns) may be misleading for some readers. Moreover, from what I learned from the rebuttal letter, it seems that the predicted signature of the $\langle L \rangle$ OAM monopole is not connected to a Berry curvature monopole in general, but it appears to be a particular feature of TaAs that these two quantities coincide close to the Weyl-point. According to the calculations mentioned in the rebuttal letter, this (accidental?) coincidence also appears for some other but not all Weyl-points. When reading the title, abstract, introduction and discussion section, I am under the impression that the $\langle L \rangle$ OAM monopole is a universal feature of Berry curvature monopoles, which does not seem to be true. These sections should be revised to clarify this point for the reader.”

Authors: We thank the Reviewer for the efforts. Below, we provide a detailed response to the points on the experimental data. We believe that these answers reinforce our experimental evidence in support of the predicted OAM winding. Moreover, we have revised the presentation of the checkerboard patterns, and now present momentum distributions that reflect the topological band (v_+).

We agree that the non-trivial OAM winding, that we find in the TaAs class, is not a universal feature of Berry curvature monopoles (BCM), as shown by our supporting calculations. However, we are convinced that this does not diminish the relevance of our findings. While the eigenstates for a Dirac-Hamiltonian universally show non-trivial winding, the underlying microscopic degrees of freedom vary from system to another. Examples include graphene (sublattice) and surface states in topological insulators (spin angular momentum). To probe the winding, the experimental probe must couple to the respective degrees of freedom. This has been achieved by quasi-particle interference STM imaging and dichroic ARPES in graphene [Nature 574, 219 (2019) and Phys. Rev. Lett. 107, 166803 (2011)] and by spin-resolved ARPES in TI [Science 323, 919 (2009)]. To the best of our knowledge, there is presently no experimental method to systematically measure the momentum-resolved Berry curvature in solids, prohibiting the “immediate detection” of BCM. Yet, in a given Weyl-semimetal system other quantities, which are experimentally accessible, may still reflect the BCM and wind non-trivially. For example, DFT calculations have predicted a non-trivial winding of the spin angular momentum (SAM) in the magnetic WSM HgCr₂Se₄ [Phys. Rev. Lett. 107, 186806 (2011)]. In TaAs we find that the OAM (but not the SAM) shows non-trivial winding, providing the possibility to use CD-ARPES as a probe because it couples to the orbital degrees of freedom. The relevance of the OAM in TaAs may be related to the fact that the energy scale associated with inversion-symmetry breaking dominates over the SOC energy scale, as our experiments and calculations also reveal.

In order to emphasize these points more clearly and to avoid possible misunderstandings we have revised the title, abstract, introduction and discussion sections.

Lastly, we emphasize that we purposefully use the term “signature” in the title and abstract as opposed to, e.g., “observation”. We believe it is common convention that the term “signature” implies considerable (but not comprehensive) experimental evidence for a theoretical prediction. Given the previous absence of any experimental information about the momentum-dependence of the Bloch wave functions in a WSM, we believe that this claim is justified for our present results.

A2 **Reviewer 2:** “Here is a detailed description of my concerns:

(1) Comments on the interpretation of the experimental data

Assuming for the time being that the calculated texture $\langle L \rangle$ OAM does reflect the texture of the Berry curvature, the experimental data presented here does not seem to agree with what is expected from the an-initio calculations for the Berry curvature monopole: a. The ab-initio calculations in Fig. 4b predict that the Berry curvature monopole consists of a band crossing of two bands with opposite OAM. This implies that, firstly, (i) at the Weyl-crossing point, there should be vanishing OAM, and secondly, (ii) to the left and to the right of Weyl-point, the upper and the lower branch of the cone should have opposite OAM. The circular dichroism (CD) data presented here does not show any signs of (i) vanishing OAM at the predicted crossing point, instead, the vicinity of the Weyl-point is completely blue with no apparent modulation of the CD-intensity. This fact contradicts the claim that a Berry curvature monopole was observed. The authors seem to imply in their rebuttal letter that this is due to momentum-dependent cross-section variations. However, I expect that the authors are aware that such matrix element effects can be suppressed in neighbouring Brillouin zones or by changing the photon energy, so unless the authors can produce such data, I am afraid that I do not find the current presentation very convincing.

The authors stress that they can show one-half of what is required for (ii), namely that the CD switches sign for an EDC line cut on the right-hand side of the predicted position of the cone at $k_x = 0.45 \text{ \AA}^{-1}$. However, this sign change is a necessary but not sufficient condition for the existence of the Weyl-point, and the inconsistencies described above seem to contradict its existence based on the data presented here.

One could imagine to shift the calculation a little bit to the right and a bit further down to align the Weyl-point with the white area between the red and blue regions. However, in that case it is unclear why there would be no signature of the red band reappearing close to the Fermi-level on the left side of the Weyl crossing, because then the energy resolution of 50 meV (as stated in the methods section) should be sufficient to resolve the splitting between the lower and the upper branch of the cone.

If the authors disagree with this assessment, it would be helpful to produce a simulated CD intensity distribution based on the ab-initio calculation that is taking into account Gaussian energy and momentum broadening and compare it to the measured CD data, rather than just plotting spaghetti plots on top of the data. I expect that such analysis would make it obvious that the experimental data does not agree very well with what is expected from the calculations.”

Authors: We thank the Reviewer for these comments and helpful suggestions.

We agree that under ideal experimental conditions the CD would be expected to reverse exactly at the Weyl point and to reverse at any k above and below the Weyl point (WP), just as the OAM in our calculations. Deviations from the ideal situation arise from (a) experimental broadening and (b) a momentum-dependent intensity variation of the band v_+ for left circularly polarized (Fig. 4e of the main draft). While the nominal energy resolution of the experiment is 50 meV, there are other sources of energy broadening: an intrinsic broadening along momentum k_z due to exponential damping of the final-state wave function into the bulk, finite angular resolution ($\sim 0.1^\circ$ corresponding to $\sim 0.02 \text{ \AA}^{-1}$), and intrinsic line-width broadening due to defects. Overall, the experimentally observed peak widths in the EDC amount to ~ 200 meV.

In order to visualize the ARPES data in an additional way, we carried out a curvature analysis of the data sets in Fig. 4d,e of the main draft. Such an analysis is established and can enhance the visibility of features with weak intensity, thus constituting a useful addition to the color-scaled intensity plots. The results are shown in Figure S10 of the supplemental material. One can see that in the curvature data set for I_L the band v_+/c_- can be traced to higher energies, essentially up to E_F , with a dispersion that is in good agreement with the calculation. This analysis clearly supports that the red/upwards dispersing and blue/downwards dispersing branches of v_+/c_- , which cross at the Weyl point, are predominantly excited by left and right circularly polarized light and thus carry opposite OAM, in agreement with the calculations.

Furthermore, as suggested by the Reviewer, we have modeled energy distribution curves (EDC) for intensities and the CD at different momenta based on the DFT-calculated band positions

and the experimentally observed broadening. Such EDC were estimated assuming (i) a constant, momentum-independent intensity and (ii) a momentum-dependent reduction of intensity for the band v_+ towards higher energies for left circularly polarized light, which is observed in Fig. 4e. Moreover, in (i) we assumed fully selective excitation (i.e. 100 % CD for a OAM-polarized bands) and in (ii) we assumed a CD of 80 % for OAM-polarized bands, as estimated from the experimental data and in agreement with previous works on TI (e.g., Phys. Rev. Lett. 110, 216801 (2013)).

The results are shown in Fig. 1 in this response letter and compared to the measured EDC. One can see that, left of the WP, the CD sign reversal above the WP is almost entirely suppressed already for the ideal scenario (i) and is fully suppressed for the case (ii). A comparably clear sign reversal is, however, visible right of the WP for both scenarios, in agreement with the experimental data. The missing sign reversal left of the WP is thus attributed to the smaller energy distance of the two bands v_+ and c_- , the experimental broadening and the momentum-dependent intensity. For the case (ii) there is also a finite CD signal at the Weyl point, originating from different intensities for the upwards and downwards dispersing parts of the band v_+ .

The above considerations indicate that the measurements do not contradict our calculations, but rather that effects of line broadening and momentum-dependent intensity variations yield certain distortions from the ideal, calculated scenario. On the other hand, salient predictions of the theory, namely the SAM polarization of the bands v_+ and v_- , the momentum-dependent OAM sign change of the band v_+ across the WP and opposite OAM polarizations of the bands v_+ and c_- (right of the WP) are clearly confirmed by our data. Focusing on the band v_+ in Fig. 4d and e of the main draft, one observes that the “red”, upwards dispersing part (right of the WP) and the “blue”, downwards dispersing part (left of the WP) are strongly selectively excited by left and right circularly polarized light, respectively. This is fully in line with the theoretical prediction assigning opposite OAM to these parts and confirms a sign change of the OAM across the Weyl point, i.e. where the dispersion changes from upwards to downwards.

We have included the Fig. 1 of this response letter and a corresponding discussion in the supplementary material (Fig. S11). We have also included another representation of the CD-ARPES data in Fig. 4c, where we plot energy positions and CD signal of the bands as obtained from fits to the measured EDC.

Lastly, we would like to emphasize that the presented experimental approach is new and challenging. CD-ARPES and spin-resolved ARPES experiments at soft X-ray energies have become possible only very recently and have, so far, only been performed on simple, metallic model systems (see e.g. Ref. [26]). Our work constitutes the first instance for such measurements on a bulk system with intrinsic spin and orbital polarization, as also emphasized by Reviewer 3. As such, the experimental conditions were carefully chosen. The grazing light incidence enables the correspondence of the CD to one specific OAM component and, thus, the possibility for a direct comparison to theory. Our data in Fig. S2 over essentially the entire soft X-ray range show that the cross section of the Weyl states for this experimental geometry is clearly optimized near the chosen photon energy of $h\nu = 590$ eV. A comparison to higher Brillouin zones at the same photon energy is not strictly possible since the k_z -plane of the W2 nodes is no longer probed in this case (cf. Fig. 1f).

A3 Reviewer 2: “b. In the main text of the manuscript and in the rebuttal letter, the authors imply that the experimentally observed and simulated checkerboard patterns in Figs. 3a-b and 4e are a signature of the Weyl-points in momentum space. As I have pointed out in the previous round of reviews - a point that has been ignored in the authors’ rebuttal letter - such a comparison appears to be misleading the reader. This is because of the large energy integration window that was used to produce these plots, which includes both trivial bands that are not forming a Weyl cone (v_-) as well as bands forming a Weyl cone (v_+, c_-). However, because the trivial v_- bands seem to have accidentally almost the same OAM texture as the bands involved in the Weyl-cone (see calculations in Fig. 4b), the large integration window may produce a checkerboard pattern that is not due to the Weyl bands but due to the contributions from the trivial v_- bands. A more suitable analysis would involve an integration window that is limited to Weyl bands and not any other bands.

Moreover, as we can see from the OAM texture of the trivial v_- band, it appears that a trivial band

FIG. 1. **Energy distribution curves (EDC) and circular dichroism (CD) close to the W_2 Weyl point.** The first two columns show modeled EDC for intensities and CD. The third column shows the corresponding experimental data for different k_x around the Weyl point, as labeled on the right (the calculated Weyl point is at $k_x = 0.51 \text{ \AA}^{-1}$). Blue and red color indicate data taken with right and left circularly polarized light. The model is based on the energy positions of v_- , v_+ and c_- obtained by DFT with a mean Gaussian peak broadening as obtained from the ARPES data (see text). In the first column a k_x -independent cross section and a full OAM selectivity is assumed, while in the second column, cross sections vary for different momenta (see text) and the OAM selectivity was set to 80%.

not forming a Weyl-point may also abruptly change the sign of its CD- intensity at a non-high symmetry point, and may therefore produce a similar checkerboard pattern. The observation of a checkerboard pattern is therefore a necessary but no sufficient condition for detecting a Berry curvature monopole. What would be more convincing is to compare iso-energy surfaces above and below the Weyl-point energy – I would expect that these should show an inverted checkerboard pattern (with respect to each other) if the Weyl-point consists of a crossing of two bands with opposite OAM.”

Authors: We agree with the Reviewer that an integration over both bands in Fig. 4 might be misleading to the reader. In the revised manuscript, we have therefore replaced the momentum maps in Fig. 4 by new maps that were obtained within a small integration window at an energy of -0.2 eV, where the experimental data reflect the band v_+ . As discussed in the previous point, the sign change above the WP can only be resolved in the data right of the WP. The OAM sign

change (right of the WP) is observed in the data in Fig. 4h, providing strong evidence that bands v_+ and c_- carry opposite OAM in agreement with the calculations. In Fig. 3 we still show data integrated over both bands, in order to illustrate the excellent correspondence of calculated OAM and measured CD over large energy and momentum ranges.

A4 Reviewer 2: “(2) Comments on the interpretation of the theoretical results

I appreciate that the authors have in their rebuttal letter clarified the difference between the OAM as conventionally defined in the literature and the quantity $\langle L \rangle$ that they also call OAM and which they use in all of their calculations presented in all of the figures of the manuscript. I believe that this distinction is absolutely crucial for the reader and needs to be mentioned in the discussion section because it is not yet clear that the quantity $\langle L \rangle$ OAM is really related to the Berry curvature for Weyl-semimetals in general, as is claimed in the first sentence of the discussion section.”

Authors: We have added a clarifying explanation in the discussion section.

A5 Reviewer 2: “What I find concerning is that the authors state in their rebuttal letter that there are many examples of other Weyl-semimetals with both type-I and type-II Weyl points that do not seem to show any non-trivial winding of the quantity $\langle L \rangle$ OAM. This seems to imply that what the authors try to probe in the experiment is actually not the intrinsic Berry curvature monopole, but a quantity that is (accidentally?) equivalent to the Berry curvature in the specific compound TaAs, but not in other compounds. I would therefore strongly recommend that the authors should modify their title to make clear that their claims are specific to TaAs and not necessarily generalizable to other Weyl-semimetals or even other topological materials, such as those hosting higher-order fermions. They should also to disclose in the discussion section that there are some Weyl-points that only show a trivial winding of the quantity $\langle L \rangle$ OAM, despite being Berry curvature monopoles. They should also comment (and show at least in the supplementary materials) if for those Weyl-points the band crossing is made up of two bands with opposite $\langle L \rangle$ OAM. This is not clear from the discussion section in its current state.”

Authors: We have revised title and discussion sections according to the suggestions of the Referee. We have also included calculations for LaAlGe in the supplementary material, showing the absence of OAM winding and the absence of a crossing of bands with opposite OAM at type-II Weyl nodes.

As we stated above, we believe that the fact that the OAM winding is not a universal feature of BCM does not diminish the relevance of the results. While the pseudospin/Berry curvature winding in a Dirac Hamiltonian is universal, the underlying microscopic degrees of freedom are not. An experimental probe has to couple to the relevant degrees of freedom. It is therefore plausible, that there is at present, to our knowledge, no universal, momentum-resolved probe of Berry curvature in solids. On the other hand, depending on the relevant degrees of freedom, other quantities, derived from the electronic wave functions, may still reflect the BCM and wind non-trivially. While not universal, these quantities may have the virtue of being accessible by an experimental probe. In the present work on the paradigmatic WSM TaAs, the OAM is such a quantity and we have used CD-ARPES to provide experimental evidence for its non-trivial winding. In magnetic WSM one may naturally expect the SAM to play a more central role. Indeed, DFT calculations indicate a non-trivial winding of the SAM in the magnetic WSM HgCr_2Se_4 [Phys. Rev. Lett. 107, 186806 (2011)]. From a fundamental theoretical perspective the usefulness of a non-universal quantity, like the OAM, might seem limited. Yet, such quantities can provide a crucial connection between theory and experiment in real materials.

AUTHORS' REPLY TO THE POINTS OF REVIEWER 3

B1 Reviewer 3: “The true experimental proof of a topological insulator should be based on measurements of its bulk properties such as the winding number. Some theoretical proposals have been made to directly measure the winding number (for example, PRB 92, 195144 (2015)), but, to my best knowledge, it has not been realized. It is the reason why people resort to detection of surface states, which should be regarded as an indirect evidence in a strict sense. It is one of the reasons why theory usually takes the leading role in studies of topological materials – experiments provide supporting evidences.

The story goes similar for Weyl semiconductors; we usually look for Fermi arcs. However, a more direct evidence should be observation chiral structure of bulk Weyl fermion states and other bulk properties. It is what Unzelm et al did in this work using bulk sensitive CD and spin-resolved ARPES for the first time. This scheme therefore can be applied to other materials in the future to establish Weyl fermion states of materials. More importantly, it will be useful in studying materials for which band calculations do not work well, i.e., correlated materials. In that case, experiments could take the leading role. The importance of this aspect is something I did not appreciate enough in the first review.

The manuscript also has useful discussion on the energy hierarchy and clearly showed OAM-based origin of the spin splitting in which the leading energy terms is determined by the OAM state rather than spin. This is the first experimental proof for such energy hierarchy in a 3D bulk system and provides a new microscopic insight on the Weyl physics.

There are some comments/questions on the replies to my previous specific questions. However, the answers to my questions/comments should not alter the main conclusion. As such, I fully support publication of the manuscript in Nature Communications. ”

Authors: We are grateful to the Reviewer for pointing out the novelty of our results and for recommending our work for publication.

B2 Reviewer 3: “Here are my comments to replies (to my specific comments in the first round) for the authors to consider.

1. The crystalline mirror plane is set as the experimental mirror plane as noted the Supplementary. Does that play a role at all in making the geometric effect negligible? Or is it solely from the fact that the final state is close to a free electron state? (Normally, the best experimental geometry would be setting the sample orientation so that the Weyl point falls on the experimental mirror plane since the geometrical contribution should be zero on the experimental mirror plane. However, this does not work in this case as OAM (L_x) also has odd distribution about the mirror plane.) ”

Authors: We thank the Reviewer for his interesting comments. While we believe that the free-electron like character is very important, further effects of the experimental geometry could play a role and will be investigated in future experiments. One important aspect in our experiments, probably not realized in most other setups, is the grazing incidence so that the photon polarization nearly aligns with the x axis. This enables the close correspondence of the CD to one specific OAM component. We agree that it would be interesting to change the geometry in such a way as to choose a highly symmetric experimental geometry with respect to the Weyl point.

B3 Reviewer 3: “2. I agree with the authors that the final state is very close to a free electron state. This is no thorough experimental/theoretical proof that final state effect is negligible with a free electron final state in spite of mounting signs in experimental results. It would be important to concretely prove it in order to establish SX CD-ARPES as a reliable tool to investigate OAM. I believe the added sentences are appropriate to describe the current situation.”

Authors: We agree with the Reviewer that our work probably only marks a first but important step towards establishing SX CD-ARPES as a reliable probe to investigate OAM. We are presently working together with experts in numerical photoemission theory to investigate in further detail the step from the observed photoemission intensities to the ground-state DFT theory.

REVIEWER COMMENTS

Reviewer #2 (Remarks to the Author):

The authors have done a good job in clarifying that the predicted non-trivial winding of the $\langle L \rangle$ OAM is not a universal feature of all Weyl-points, but that it does seem to correspond to the winding of Berry curvature around the type-1 Weyl points in TaAs. I also agree with the authors that this non-universality does not diminish the scientific merit of their claims, because it offers a new route to experimentally accessing Berry curvature monopoles (in at least some Weyl-semimetals).

In my previous report, I voiced concerns about inconsistencies between the experimental data and the calculated predictions for the $\langle L \rangle$ OAM texture presented in Fig. 4. This figure is particularly important because it supposedly backs the central claim of the paper of observing signatures of a Berry curvature monopole by observing a crossing of two bands with opposite OAM. The authors responded to these concerns by extending the analysis of their existing data, which includes

1. A simulation of the expected EDCs based on the ab-initio calculations in Fig. S11 where they assume an intensity modulation, a finite strength of the circular dichroism asymmetry, and experimental broadening. By fine-tuning these parameters, they explain that two of the expected signatures of the Berry curvature monopole should disappear in the experiment (the vanishing of OAM at the Weyl point, and the reversal of the OAM for the upper and lower part of the cone to the left and right of the Weyl point), which could explain why they are not observed experimentally
2. A curvature analysis presented in Fig. S10 which they interpret (in conjunction with Fig. 4e) as evidence for an intensity modulation of the red band (in the ab-initio calculations) close to the Fermi-level
3. An additional Fig. 4c that shows the results of some EDC fits
4. A new Fig. 4f that shows an OAM checkerboard pattern of an iso-energy surface below the Weyl-point, which appears to be in agreement with the ab-initio prediction

I have the following comments/questions about these new pieces of analysis

1. The authors used a model with multiple tuning parameters (including strong broadening and intensity modulation) to explain the lack of experimentally observable signatures for the $\langle L \rangle$ OAM monopole. Whilst this model is motivated reasonably well, the large number of parameters is concerning since one may wonder whether any kind of experimental data may be reproduced with such a model. It would be much more convincing if the authors were able to present data without intensity modulation and with reduced broadening, which could resolve the two expected signatures of the Berry curvature monopole mentioned above. However, the authors seem unwilling or unable to improve the quality of their experimental data with additional experiments, for instance by probing the bands in a neighboring Brillouin zone (which may also require changing the photon energy a bit), which could suppress such intensity modulation. Whilst I find this unfortunate, I can also understand that during the Covid-19 pandemic the opportunities for additional experiments are limited.
2. Fig. S10b shows not just a downwards dispersing band $v+$, but also contributions from an upwards dispersing band right at the Fermi-level. This upwards dispersing band might be the band labelled $c-$ in Fig. S10a. If that is the case, it is not clear that the band labelled as $v+$ really extends to the Fermi-level. If the authors disagree, then how do they explain the strange kink at around $E_b \sim 10-30$ meV in Fig. S10b?
3. I am confused by the red dots shown in Fig. 4c in the range of $k_x = -0.7$ to -0.5 \AA^{-1} and $E \sim 100$ meV. I cannot see any intensity in this range in Fig. 4e, and one would not expect any intensity there based on the ab-initio calculations. More generally, I do not think that this new Figure 4c helps to convince the reader that theory and experiment agree well. The authors may instead consider providing an E vs k pseudocolor plot of the predicted CD spectrum over the same range as Fig. 4b

based on the model that they used to produce the middle column of Fig. S11

4. Whilst the agreement between theory and experiment is encouraging, I am not sure that the new fig. 4f is a clear signature of the presence of Berry curvature monopoles. It may be helpful to plot an additional iso-energy surface at the Fermi-energy for a comparison. From the ab-initio calculations I would expect that the checkerboard pattern should be inverted compared to fig 4f, which may be a more direct signature of a band crossing of two bands with opposite OAM.

Whilst I remain concerned about the quality of the experimental data backing the main claim of the manuscript (and thus the strength of the experimental evidence), I am inclined to recommend publication of a revised version that is taking into account the feedback given above. This is because I think that the overall conceptual novelty of the paper is of interest for the broad audience of Nature Communications.

AUTHORS' REPLY TO THE POINTS OF REVIEWER 2

A1 **Reviewer 2:** “The authors have done a good job in clarifying that the predicted non-trivial winding of the $\langle L \rangle$ OAM is not a universal feature of all Weyl-points, but that it does seem to correspond to the winding of Berry curvature around the type-1 Weyl points in TaAs. I also agree with the authors that this non-universality does not diminish the scientific merit of their claims, because it offers a new route to experimentally accessing Berry curvature monopoles (in at least some Weyl-semimetals).

In my previous report, I voiced concerns about inconsistencies between the experimental data and the calculated predictions for the $\langle L \rangle$ OAM texture presented in Fig. 4. This figure is particularly important because it supposedly backs the central claim of the paper of observing signatures of a Berry curvature monopole by observing a crossing of two bands with opposite OAM. The authors responded to these concerns by extending the analysis of their existing data, which includes

1. A simulation of the expected EDCs based on the ab-initio calculations in Fig. S11 where they assume an intensity modulation, a finite strength of the circular dichroism asymmetry, and experimental broadening. By fine-tuning these parameters, they explain that two of the expected signatures of the Berry curvature monopole should disappear in the experiment (the vanishing of OAM at the Weyl point, and the reversal of the OAM for the upper and lower part of the cone to the left and right of the Weyl point), which could explain why they are not observed experimentally
2. A curvature analysis presented in Fig. S10 which they interpret (in conjunction with Fig. 4e) as evidence for an intensity modulation of the red band (in the ab-initio calculations) close to the Fermi-level
3. An additional Fig. 4c that shows the results of some EDC fits
4. A new Fig. 4f that shows an OAM checkerboard pattern of an iso-energy surface below the Weyl-point, which appears to be in agreement with the ab-initio prediction”

Authors: We thank the Reviewer for the additional efforts and for the positive comments regarding our clarifications in the previous response.

A2 **Reviewer 2:** “I have the following comments/questions about these new pieces of analysis 1. The authors used a model with multiple tuning parameters (including strong broadening and intensity modulation) to explain the lack of experimentally observable signatures for the $\langle L \rangle$ OAM monopole. Whilst this model is motivated reasonably well, the large number of parameters is concerning since one may wonder whether any kind of experimental data may be reproduced with such a model. It would be much more convincing if the authors were able to present data without intensity modulation and with reduced broadening, which could resolve the two expected signatures of the Berry curvature monopole mentioned above. However, the authors seem unwilling or unable to improve the quality of their experimental data with additional experiments, for instance by probing the bands in a neighboring Brillouin zone (which may also require changing the photon energy a bit), which could suppress such intensity modulation. Whilst I find this unfortunate, I can also understand that during the Covid-19 pandemic the opportunities for additional experiments are limited.”

Authors: We appreciate that the Reviewer estimates our model as reasonably well motivated. As discussed in detail in the previous response, the model is intended to illustrate how experimental broadening and a momentum-dependent intensity variation yield deviations from an ideal scenario in the CD pattern close to the Weyl points. Although the model contains a number of parameters, most of them are fixed either by our DFT calculation or by the experimental spectra. The peak positions are fixed by the DFT calculation and the line widths are fixed by the experimentally observed value. Moreover, the signs of the dichroism are fixed by the signs of the OAM in the DFT calculation. To illustrate the effect just of the experimental broadening we consider the case (i). Case (i) shows that, given the experimental broadening of 200 meV, even under otherwise “ideal” conditions (no momentum-dependent intensity variation and a CD of $\pm 100\%$ for an OAM-polarized band), the CD sign switch left of the Weyl point would show up only as faint feature. Case (ii) then shows that under more realistic conditions this CD switch left of the Weyl point disappears, while it remains robust right of the Weyl point.

We agree with the Referee that future experiments with further optimized conditions might be able to resolve the OAM/CD reversal above the Weyl point, although, as explained in the previous response, the experimental parameters were chosen very carefully. Nevertheless, our observations

of CD sign reversals for the lower part of the Weyl cone (band v_+) and between upper and lower Weyl cone on the right side of the Weyl point (bands v_+ and c_-) provide strong evidence that the Weyl point constitutes a crossing point of bands with opposite OAM. We also point out that the experimental methods (spin-resolved and CD-SX-ARPES) are applied here for the first time to study spin- and orbital-polarizations in a bulk system. One may compare our data to previous SX-ARPES results on TaAs *without* CD/spin analysis to estimate that our present data are of excellent quality [see Figs. 3f,g in Nat. Phys. 11, 724–727 (2015) showing DFT/SX-ARPES cuts analogous to our cuts in Figs. 4a-d. Note the different nomenclature of the Weyl points in this reference: The W_2 points in our work are labeled “ W_1 ” in the reference.].

- A3 **Reviewer 2:** “2. Fig. S10b shows not just a downwards dispersing band v_+ , but also contributions from an upwards dispersing band right at the Fermi-level. This upwards dispersing band might be the band labelled c_- in Fig. S10a. If that is the case, it is not clear that the band labelled as v_+ really extends to the Fermi-level. If the authors disagree, then how do they explain the strange kink at around Eb 10-30 meV in Fig. S10b?”

Authors: We agree with the Reviewer that also in the I_L data set one can discern traces of the band c_- . In agreement with the direct intensity data, the spectral weight of this feature is, however, more pronounced in the I_R data set, indicating an opposite CD when compared to the band v_+ at the same momentum (right of the Weyl point). Despite these signatures of the band c_- it is possible to trace the band v_+ to higher energies in the curvature plots than in the intensity plots, which is the statement we make in the supplement. To be more specific we added the following statement in the supplemental material in the discussion of Fig. S10: “*While the curvature analysis allows one to trace the band v_+ reasonably well up to the Weyl point, additional signatures of the band c_- complicate a clear assignment above the Weyl point.*”

- A4 **Reviewer 2:** “3. I am confused by the red dots shown in Fig. 4c in the range of $kx=-0.7$ to -0.5 \AA^{-1} and E 100 meV. I cannot see any intensity in this range in Fig. 4e, and one would not expect any intensity there based on the ab-initio calculations. More generally, I do not think that this new Figure 4c helps to convince the reader that theory and experiment agree well. The authors may instead consider providing an E vs k pseudocolor plot of the predicted CD spectrum over the same range as Fig. 4b based on the model that they used to produce the middle column of Fig. S11.”

Authors: While the intensity corresponding to the red dots is difficult to discern in the false-color plot in Fig. 4c, it can be seen clearly in the corresponding EDC. Two examples are shown in Fig. S11. The dots in Fig. 4c are obtained from fits to EDC. We believe that this plot provides a useful addition to the other visualizations of the data in Fig. 4, and we therefore decided to keep it in the main manuscript. Following the suggestion of the Reviewer we have added pseudocolor plots of cases (i) and (ii) of the model in the supplemental material as Fig. S12. Moreover, we added the following statement in the main text referring to Fig. 4c: “*It is noteworthy that the magnitude of the CD in general does not reach 100 %, as seen, e.g., from the analysis in Fig. 4c.*”

- A5 **Reviewer 2:** “4. Whilst the agreement between theory and experiment is encouraging, I am not sure that the new fig. 4f is a clear signature of the presence of Berry curvature monopoles. It may be helpful to plot an additional iso-energy surface at the Fermi-energy for a comparison. From the ab-initio calculations I would expect that the checkerboard pattern should be inverted compared to fig 4f, which may be a more direct signature of a band crossing of two bands with opposite OAM.”

Authors: We agree with the Reviewer that an observation of a reversal of the CD momentum pattern below and above the Weyl point would be desirable. For reasons discussed in point 1 and in the previous response our present data do not allow us to resolve this sign reversal over the full momentum range. To address this point, we have added the following statement in the main text: “*In particular, unlike for the band v_+ , our present data does not allow us to discern a CD reversal of the band c_- across the Weyl point close to the Fermi level.*”

Note, however, that right of the Weyl point our data clearly confirm an opposite OAM for the bands v_+ and c_- (Fig. 4h), in part due to the larger energy separation of these bands at these momenta. Therefore, our data confirm a reversal of the OAM of the band v_+ across the Weyl point and an opposite OAM of bands v_+ and c_- at momenta right of the Weyl point.

A6 **Reviewer 2:** “Whilst I remain concerned about the quality of the experimental data backing the main claim of the manuscript (and thus the strength of the experimental evidence), I am inclined to recommend publication of a revised version that is taking into account the feedback given above. This is because I think that the overall conceptual novelty of the paper is of interest for the broad audience of Nature Communications.”

Authors: We thank the Reviewer 2 for acknowledging the novelty of our results and hope that the feedback has been taken into account appropriately.